# *Plasmodium falciparum* ligand binding to erythrocytes induce alterations in deformability essential for invasion

Xavier Sisquella[1], Thomas Nebl[1], Jennifer K Thompson[1], Lachlan Whitehead[1], Brian M Malpede[2,3], Nichole D Salinas[2,3], Kelly Rogers[1], Niraj H Tolia[2,3], Andrea Fleig[4], Joseph O'Neill[1], Wai-Hong Tham[1,5], F David Horgen[6], Alan F Cowman[1,5]*

[1]The Walter and Eliza Hall Institute of Medical Research, Parkville, Australia; [2]Molecular Microbiology and Microbial Pathogenesis, Washington University School of Medicine, St. Louis, United States; [3]Biochemistry and Molecular Biophysics, Washington University School of Medicine, St. Louis, United States; [4]The Queen's Medical Center and John A. Burns School of Medicine, University of Hawaii, Honolulu, United States; [5]Department of Medical Biology, The University of Melbourne, Parkville, Australia; [6]Department of Natural Sciences, Hawaii Pacific University, Kaneohe, United States

**Abstract** The most lethal form of malaria in humans is caused by *Plasmodium falciparum*. These parasites invade erythrocytes, a complex process involving multiple ligand-receptor interactions. The parasite makes initial contact with the erythrocyte followed by dramatic deformations linked to the function of the Erythrocyte binding antigen family and *P. falciparum* reticulocyte binding-like families. We show EBA-175 mediates substantial changes in the deformability of erythrocytes by binding to glycophorin A and activating a phosphorylation cascade that includes erythrocyte cytoskeletal proteins resulting in changes in the viscoelastic properties of the host cell. TRPM7 kinase inhibitors FTY720 and waixenicin A block the changes in the deformability of erythrocytes and inhibit merozoite invasion by directly inhibiting the phosphorylation cascade. Therefore, binding of *P. falciparum* parasites to the erythrocyte directly activate a signaling pathway through a phosphorylation cascade and this alters the viscoelastic properties of the host membrane conditioning it for successful invasion.

*For correspondence: cowman@wehi.edu.au

Competing interests: The authors declare that no competing interests exist.

## Introduction

Malaria is a major global disease of humans and the most severe form is caused by *Plasmodium falciparum*. This protozoan parasite has a complex life cycle, however, the symptoms of malaria are mediated by the asexual blood stage that is initiated by the entry of the merozoite form into the host erythrocyte. During the initial steps of the invasion, the merozoite intermittently contacts the erythrocyte until it attaches and initiates internalization. Attachment and internalization involve a series of dramatic changes that include deformation of the erythrocyte (*Figure 1A*) and calcium (Ca²⁺) influx after which a tight junction is formed and invasion mediated using force generated by the parasite actomyosin motor (*Weiss et al., 2015*; *Volz et al., 2016*). The mechanical alterations of the erythrocyte that occur during merozoite invasion have been described using video-microscopy (*Weiss et al., 2015*; *Gilson et al., 2009*). Merozoite invasion involves the interaction of multiple parasite ligands with specific erythrocyte receptors that include merozoite surface proteins (MSPs)

**Figure 1.** *P. falciparum* EBA-175 RII binding to GPA increases deformability of the erythrocyte. (**A**) Live imaging time frames showing a merozoite deforming an erythrocyte. Scale bar 5 µm. (**B**) Atomic force microscopy (AFM) screen of the effect of *P. falciparum* invasion ligands on the erythrocyte Young's modulus (**E**). (**C**) Schematics showing EBA-175 and EBA-140 domain structure (top). The bottom panel is the AFM Young's modulus of erythrocytes treated with EBA-175 region II, region III-IV and EBA-175 RII in the presence of EBA-175 RII antibodies (EBA-175+Ab). (**D**) AFM Young's

*Figure 1 continued on next page*

*Figure 1 continued*

modulus of neuraminidase treated erythrocytes in the presence or absence of EBA-175 RII. (**E**) EBA-175 titration on erythrocytes and comparison of the AFM Young's modulus (left) with the elongation index measured by rheology (right). 4 µg (**B**), 3.5 µg (**C**), 3.8 µg (**D**) and 1–4 µg (**E**) were added to 500 µL erythrocytes in RPMI-HEPES at 2% haematocrit. Error bars represent the mean and SEM for three independent experiments.

The following figure supplements are available for figure 1:

**Figure supplement 1.** Binding of *P. falciparum* recombinant ligands to human erythrocytes.

**Figure supplement 2.** Binding of recombinant EBA-140 RII to human erythrocytes.

(*Hodder et al., 2012*), erythrocyte binding antigens (EBAs) (*Adams et al., 2001*) and reticulocyte binding-like homologs (PfRhs) (*Rayner et al., 2000*; *Triglia et al., 2001*; *Hayton et al., 2008*; *Baum et al., 2009*), and the functional requirement of these during specific steps of the invasion has been defined (*Weiss et al., 2015*; *Volz et al., 2016*). This includes binding of the merozoite surface protein Duffy Binding-Like 1 and 2 (MSPDBL1 and 2) with unknown receptors (*Hodder et al., 2010*), EBA-175 with glycophorin A (GPA) (*Orlandi et al., 1992*), EBA-140 (also known as BAEBL) with glycophorin C (GPC) (*Maier et al., 2003*), PfRh4 with Complement Receptor 1 (CR1) (*Tham et al., 2010*) and PfRh5 with basigin (*Crosnier et al., 2011*). Nothing is known with respect to alterations that these ligand-receptor interactions mediate on the host cell during these initial stages of merozoite invasion.

Erythrocytes are very flexible and dynamic cells that are able to flow smoothly through the microvasculature and pass swiftly through the spleen. The shear elastic properties of the erythrocyte are predominantly determined by the underlying spectrin network as well as the connection of integral membrane proteins with this cytoskeleton. The erythrocyte can undergo repeated large deformations to facilitate movement through microcapillaries, and these deformations involve the dynamic remodeling of the spectrin network (*Li et al., 2007*). Additionally, under normal physiological conditions, calcium ($Ca^{2+}$) influx or treatment with certain amphipathic drugs can induce membrane budding (*Zuccala et al., 2011*; *Allan et al., 1976*; *Ben-Bassat et al., 1972*). Also, active ATP-dependent cytoskeleton forces that are uncorrelated with Brownian noise have been detected in erythrocytes (*Rodríguez-García et al., 2015*). Environmental factors can trigger post-translational modifications and change the erythrocyte membrane properties, and antibody ligation of CR1 increases erythrocyte membrane deformability (*Glodek et al., 2010*). Phosphorylation and dephosphorylation of membrane and cytoskeletal proteins is a likely mechanism by which properties of the erythrocyte membrane are regulated (*Mohandas and Gallagher, 2008*), and increased phosphorylation of erythrocyte proteins occurs on attachment of *P. falciparum* merozoites suggesting changes to the host cell cytoskeleton may be important for parasite entry (*Zuccala et al., 2016*).

In this study, we show that *P. falciparum* ligand-receptor interactions affect the deformability of the erythrocyte. In particular, EBA-175 binding to GPA causes substantial changes in the deformability of erythrocytes and activates a phosphorylation cascade that alters the viscoelastic properties of the host membrane, a process that is essential for successful parasite invasion.

## Results

### Binding of *P. falciparum* ligands to human erythrocytes affects deformability

*P. falciparum* merozoites significantly deform the erythrocyte during invasion and potentially alter the visco-elastic properties of the host cell (*Figure 1A*). The effect of *P. falciparum* ligand-receptor interactions on the visco-elastic properties of the erythrocyte was determined with recombinant proteins that bind to specific receptors. MSPDBL1, MSPDBL2 (*Hodder et al., 2010*), EBA-175 region II (EBA-175 RII), PfRh4 (*Tham et al., 2010*), PfRh5 (*Guz et al., 2014*), EBA-140 region II (EBA140 RII) (*Maier et al., 2003*) were incubated with human erythrocytes and the Young's modulus (E) calculated from the atomic force microscopy (AFM) cantilever deflection as a measure of cell stiffness (*Figure 1B*) (*Guz et al., 2014*). The recombinant proteins MSPDBL1 (*Lin et al., 2016*,

*2014*), MSPDBL2 (*Lin et al., 2016*, *2014*), PfRh4, PfRh5, EBA-140 RII and EBA-175 RII specifically bind to erythrocytes. Erythrocyte stiffness was generally reduced by binding of MSPDBL1, MSPDBL2, PfRh4, PfRh5 and EBA-140 RII (p<0.0001) with EBA-175 RII binding showing a dramatic reduction of Young's modulus (*Figure 1B*; *Figure 1—figure supplement 1* and *Figure 1—figure supplement 2*). The erythrocyte-binding domain of EBA-175 is region II, a 616 amino acid fragment consisting of two cysteine-rich Duffy binding-like (DBL) domains (F1 and F2) (*Tolia et al., 2005*), that binds GPA (*Figure 1C*) (*Sim et al., 1994*; *Salinas et al., 2014*; *Salinas and Tolia, 2014*). EBA-175 dimerizes upon receptor engagement (*Tolia et al., 2005*), and neutralizing antibodies block the dimer interface and receptor binding residues of EBA-175 RII (*Chen et al., 2013*). EBA-175 and EBA-140 contain the conserved domain architecture of the EBL family and bind similarly to GPA and GPC respectively, although the latter binds as a monomer (*Malpede and Tolia, 2014*; *Paing and Tolia, 2014*).

As controls we tested the effect of the intrinsically disordered EBA-175 III-V domain (*Blanc et al., 2014*), which does not bind erythrocytes (*Healer et al., 2013*), and it did not significantly influence erythrocyte deformability (p<0.05, *Figure 1C*). Erythrocytes incubated with EBA-175 RII together with anti-EBA-175 RII antibodies, that block binding to GPA and merozoite invasion (*Chen et al., 2013*; *Healer et al., 2013*; *Sim et al., 2011*), showed no detectable change in host cell stiffness (p>0.05) consistent with them blocking ligand binding (*Figure 1C*). EBA-175 and EBA-140 binding are both dependent on sialic acid moieties on glycophorin receptors (*Adams et al., 2001*) and thus sensitive to neuraminidase-treatment of erythrocytes. EBA-175 RII did not affect the stiffness of neuraminidase-treated erythrocytes (*Figure 1D*) indicating a direct link between EBA-175 binding to GPA and changes in erythrocyte deformability. EBA-175 RII binding showed an expected dose response on deformability, which was confirmed using ektacytometry (*Moyer et al., 2010*), consistent with the quantification of EBA-175 RII binding to erythrocytes (*Figure 1E*; *Figure 1—figure supplement 1*). Bulk rheology measures of the elongation index showed an increase in erythrocyte deformability with EBA-175 RII reaching a plateau after 35 nM (*Figure 1E*), which was close to saturation of GPA binding sites (*Poole, 2000*). Therefore, binding of EBA-175 RII to glycophorin A on the erythrocyte surface was responsible for the changes observed in deformability of the host cell.

## Binding of EBA-175 to GPA on the erythrocyte increases phosphorylation of the cytoskeleton

We next determined if the mechanism of EBA-175 RII induced deformability of erythrocytes was a result of changes in phosphorylation of cytoskeleton components. The EBA-175 RII- treated erythrocytes were radiolabeled with $^{32}$P inorganic phosphate, and ghosts containing membrane and cytoskeleton proteins extracted and analyzed by 2-dimensional (2-D) gel electrophoresis (*Figure 2A and B*). An altered intensity of $^{32}$P labeling in the EBA-175 RII treated erythrocytes indicated an overall increase of phosphorylation, and false-color overlays of aligned autoradiographs showed a shift of particular 2-D spots, revealing some proteins were multiply phosphorylated (*Figure 2A and B*; *Figure 2—figure supplement 1*). Twenty eight of the most prominent $^{32}$P-labeled protein spots were excised from an aligned preparative 2-D gel and analyzed by LC-MS/MS. This identified a subset of phosphorylated proteins corresponding to erythrocyte membrane skeleton proteins including tropomodulin-1, adducin-2, tropomyosin, beta-actin, ankyrin-1, protein 4.1 and cofilin-1. The identity of these proteins was validated by immuno-blot (*Figure 2*; *Figure 2—figure supplement 1*). 2-D image analysis of $^{32}$P autoradiographs and immuno-blots of the same membranes indicated that $^{32}$P labeled 2-D phospho-spots matched accurately with bands for tropomodulin-1 and adducin-2, with clear changes in their phosphorylation state (*Figure 2C–F*).

To provide a more quantitative approach we used dimethyl labeling and quantitative liquid chromatography-tandem mass spectrometry (LC-MSMS) to enable identification of phosphopeptides at a global level (*Boersema et al., 2010*) (*Figure 2G–H*). EBA-175 RII treated and untreated erythrocytes were digested prior to labeling of the lysine and N-terminal residues with either a light or a heavy dimethyl isotope. Global phosphoproteomics analysis of the phosphopeptide enriched Heavy (H) + Light (L) mixture revealed a high number (982) of unique phosphopeptides, and the H/L ratio of unique peptides singly, doubly or triply phosphorylated indicated an overall increase in phosphorylation in the EBA-175 RII treated samples (*Figure 2G*; *Figure 2—figure supplement 2*). Quantitative LC-MSMS analysis identified approximately 400 erythrocyte phosphoproteins and revealed a significant (p<0.05) enrichment of phosphopeptides corresponding to trans-membrane spanning

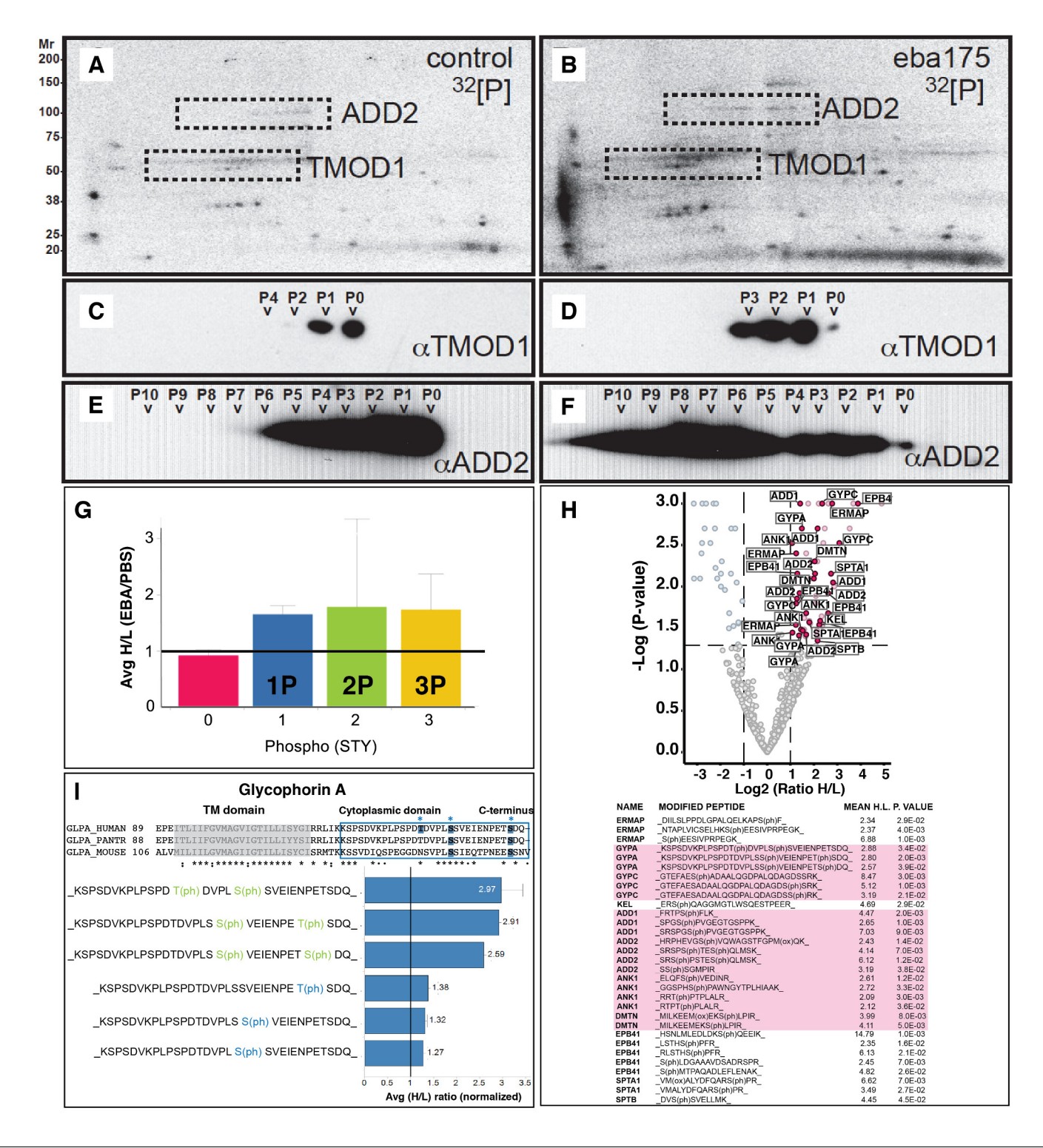

**Figure 2.** EBA-175 RII induces an increase in phosphorylation of trans-membrane and cytoskeletal erythrocyte proteins. (A–B) MW vs. pI 2D electrophoresis gel autoradiographs of EBA-175 RII treated (B) or untreated (A) $^{32}$P radio-labeled erythrocyte ghosts. (C–F) Western blot phosphorylation validation of tropomodulin 1 (C–D) and adducin-2 (E–F). P0-P10 indicate the number of phosphorylation sites. (G) Mass spectrometry phosphopeptide heavy (H) to light (L) ratios of EBA-175 (H) mixed with PBS (L) treated ghosts. (H). Volcano plot showing quantitative analysis of unique phosphopeptides. Red dots indicate significant upregulated unique phosphopeptides corresponding to cytoskeletal protein IDs highlighted in the list.

*Figure 2 continued on next page*

*Figure 2 continued*

(I) Amino acid sequence of glycophorin A (top), red square highlights the cytoplasmic domain conserved across different species. Glycophorin A unique phosphopeptides detected (bottom), red circles point out phosphorylation sites all located in the cytoplasmic domain.

The following figure supplements are available for figure 2:

**Figure supplement 1.** Binding of EBA-175 to human erythrocytes activates increased phosphorylation of the host cytoskeleton.

**Figure supplement 2.** Quantitation and identification of peptides phosphorylated after EBA-175 binding to human erthrocytes.

glycoproteins and cytoskeletal proteins that include GPA and GPC, adducin, ankyrin and dematin, with many of them multiply phosphorylated (*Figure 2H*; *Figure 2—figure supplement 2*; *Supplementary file 1* and *2*). Interestingly, both GPA and GPC phosphorylation sites were detected in the cytosolic domains of these proteins (*Figure 2I*; *Figure 2—figure supplement 2*). Therefore EBA-175 RII binding to erythrocytes induced an overall change in phosphorylation of the cells cytoskeletal proteins. These observations suggested an induced modification of the cytoskeleton and consequent increase deformability upon the specific interaction of merozoite ligands with the host erythrocyte.

## EBA-175 binding to GPA triggers a phosphorylation cascade of erythrocyte membrane glycoproteins and cytoskeleton proteins by TRPM7

To identify the kinase(s) involved in erythrocyte phosphorylation triggered by EBA-175 RII binding to GPA we tested erythrocyte kinase inhibitors that influence erythrocyte deformability. A protein kinase C (PKC) inhibitor, Gö9676 (*Bailey et al., 2014*), four transient receptor potential cation channel (TRPM7) inhibitors, FTY720, sphingosine (*Qin et al., 2013*), waixenicin A (*Zierler et al., 2011*) and NS8593 (*Chubanov et al., 2012*), three Rho-kinase inhibitors, Y-27632 (*Ruef et al., 2011*), simvastatin (*Clapp et al., 2013*) and HA1077-fasudil (*Tiftik et al., 2014*), an adenylyl cyclase and protein kinase A (PKA) stimulator, forskolin (*Muravyov and Tikhomirova, 2013*), an AMP-activated PK inhibitor, dorsomorphin (*Liu et al., 2014*), a phosphodiesterase inhibitor, isobutyl-methyl-xantine (IBMX, [*Muravyov and Tikhomirova, 2013*]), a spleen tyrosine kinase (syk) inhibitor, BAY-3606 (*Norman, 2014*), a $Ca^{2+}$ channel inhibitor, verapamil (*Ruef et al., 2011*) and three inhibitors of mechanically activated currents through channels such as TRPM7 or Piezo1, gadolinium (III), ruthenium red (*Drew et al., 2002*) and GsMTx-4 (*Dorovkov et al., 2008*). Some inhibitors had no effect on overall growth of *P. falciparum* whilst others reduced it to less than 50% compared to controls (sphingosine, Y-27632, IBMX, BAY61-3606, gadolinium (III) and verapamil) (*Figure 3A*). Strikingly, FTY720, waixenicin A and NS8593, which are TRPM7 inhibitors, blocked parasite growth completely. TRPM7 is a two-domain protein expressed in most tissues containing both a TRP ion channel and an alpha-kinase domain (*Ryazanova et al., 2004*). TRPM7 phosphorylates tropomodulin-1 at the N- terminal Ser2 and Thr54 residues (*Dorovkov et al., 2008*), and Ser2 is phosphorylated in erythrocytes (*Weber et al., 1994*). These data along with the phosphorylation of tropomodulin-1, induced by EBA-175 binding to glycophorin A (*Figure 2C and D*), suggested that TRPM7 may be involved in this phosphorylation cascade and that TRPM7 inhibitors could block parasite growth by impairing invasion.

## Increased deformability of the host erythrocyte is required for merozoite invasion

To determine if TRPM7 inhibitors, that blocked *P. falciparum* growth, directly impaired merozoite invasion we used the reversible TRPM7 inhibitor FTY720 (*Chubanov et al., 2012*). Purified merozoites were added to FTY720 or sphingosine treated erythrocytes, and invasion quantified (*Figure 3B*). FTY720 inhibited merozoite invasion ($IC_{50}$ of $34.1 \pm 1.1$ μM) whereas sphingosine had no effect. Additionally, we performed growth inhibition assays to confirm the activity of FTY720. Synchronized 3D7 trophozoites were treated with different concentrations of each compound and the parasitemia determined to obtain an inhibitory dose response. FTY720 ($IC_{50}$ $107.3 \pm 1.0$ μM) showed

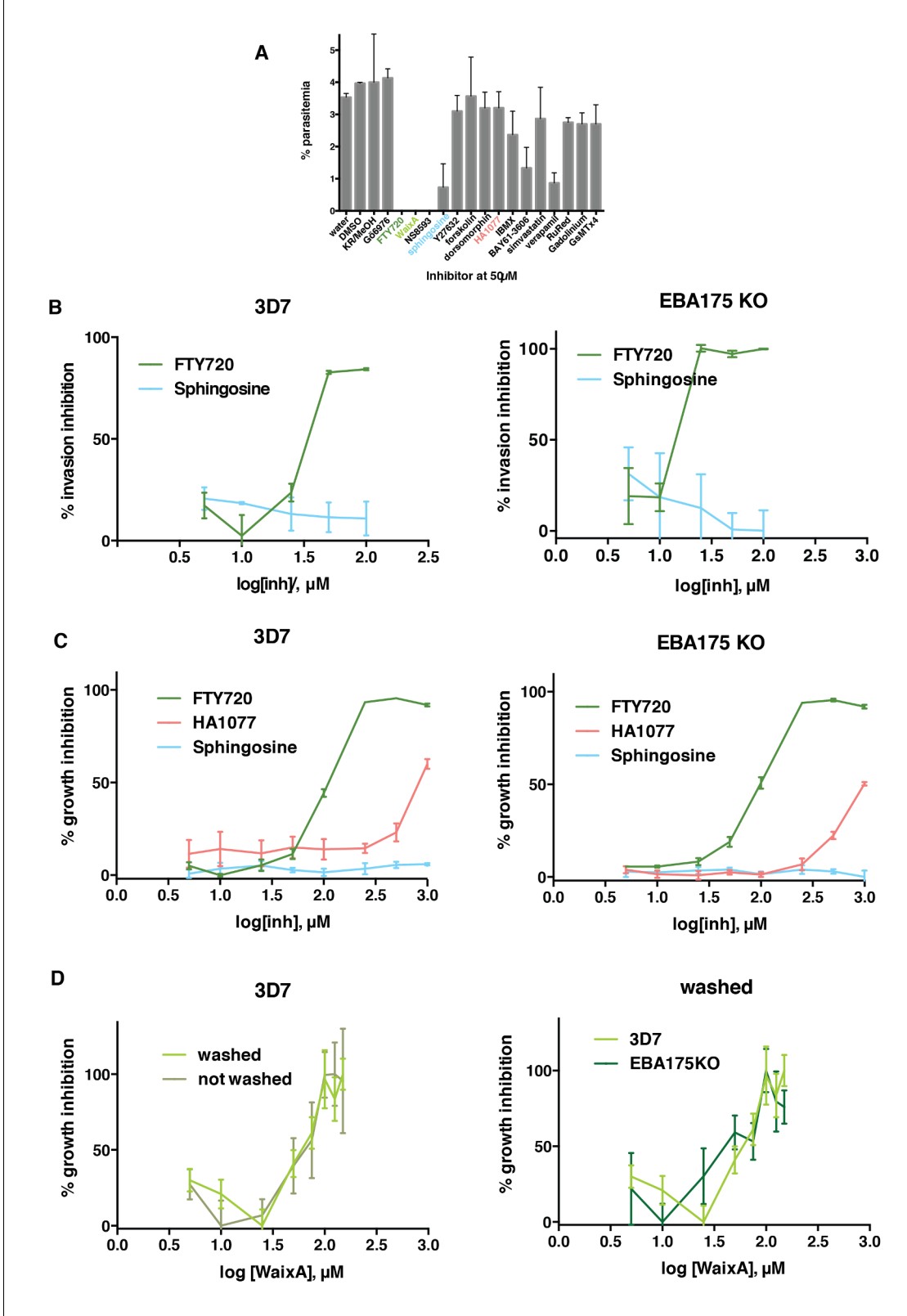

**Figure 3.** FTY720 and Waixenicin A, inhibitors of TRPM7, block *P. falciparum* erythrocyte invasion. (A) Parasitemia after 12–16 hr post addition of synchronous schizonts to erythrocytes treated with different inhibitors at 50 μM. Parasitaemia was normalized to the control. Error bars show SEM corresponding to three independent experiments. (B) Invasion inhibition assays. Purified merozoites from either 3D7 (left) or 3D7ΔEBA-175 (right) parasite lines were added to erythrocytes treated with varied concentrations of FTY720 or sphingosine. Parasitaemia was counted after 24 hr,
*Figure 3 continued on next page*

Figure 3 continued

normalized to the control and expressed as % inhibition. (C) Growth inhibition assays (GIAs) showing inhibitory activity of FTY720, HA1077 and sphingosine for 3D7 (left) and 3D7ΔEBA-175 (right) parasite lines. (D) GIAs after washing or not washing waixenicin A treated 3D7 (left) and 3D7ΔEBA-175 parasites (right). All trophozoite-synchronized cultures were treated with serial inhibitor concentrations and parasitaemia was counted after 24 hr. Error bars show SEM corresponding to three independent experiments.

potent inhibition (*Figure 3C*) whilst HA1077 and sphingosine showed little to none. To further test whether FTY720 was inhibiting invasion rather than interfering with parasite development we tested the inhibitory activity of waixenicin A (*Figure 3D*). This compound is a slow acting inhibitor of TRPM7 and therefore could be incubated with erythrocytes and washed out before testing merozoite invasion (*Zierler et al., 2011*). When waixenicin A-treated erythrocytes were tested a similar result was obtained for either washed (IC$_{50}$ 59.4 ± 1.2 µM) or unwashed (IC$_{50}$58.8 ± 1.1 µM) host cells (*Figure 3D*). Therefore, FTY720 and waixenicin A block invasion *of P. falciparum* merozoites by inhibiting an essential erythrocyte function(s).

Interestingly, the same results were observed for 3D7ΔEBA-175 parasites that lack expression of EBA-175, where FTY720 inhibited merozoite invasion (IC$_{50}$ 20.8 ± 1.2) (*Figure 3B*) and parasite growth (IC$_{50}$95.2 ± 1.0 µM) (*Figure 3C*). Similarly, waixenicin A also inhibited 3D7ΔEBA-175 growth (IC$_{50}$ 43.8 ± 1.3 µM) (*Figure 3D*). The ability of FTY720 and waixenicin A to inhibit invasion of EBA-175 deficient parasites were consistent with this ligand not being essential for merozoite invasion as other members of the EBA and PfRh family of proteins have overlapping functions (*Duraisingh et al., 2003*; *Stubbs et al., 2005*; *Lopaticki et al., 2011*). EBA-175 function is redundant but the overall function of the EBA and PfRh families is essential (*Lopaticki et al., 2011*). Whilst EBA-175 RII binding had a significant effect on erythrocyte deformability other ligands such as EBA-140 RII, which binds to GPC (*Maier et al., 2003*), and PfRh4, which binds to CR1 (*Tham et al., 2010*), also had an effect although not to the same magnitude as EBA-175 RII (*Figure 1B*). GPA is the most abundant integral protein on the erythrocyte surface, present at 10$^6$ copies per cell (*Poole, 2000*). In contrast, there are approximately 225,000 molecules of GPC per erythrocyte (*Smythe et al., 1994*), and the normal level of CR1 in Europeans is between 50–1200 molecules per cell (*Wilson et al., 1986*). The higher abundance of GPA explains the lesser effect of EBA-140 RII and PfRh4 on the deformability of treated erythrocytes. Therefore, it is likely that in the absence of EBA-175 expression other members of the EBA and PfRh protein families perform an identical function that alters the deformability of the erythrocyte through the TRPM7 pathway, and this is essential for merozoite invasion.

Binding of EBA-175 to GPA on erythrocytes altered their deformability and, in addition, induced phosphorylation of host trans-membrane spanning glycoproteins and cytoskeletal proteins. Inhibitors of TRPM7, a divalent cation channel with a kinase domain, blocked merozoite invasion. The hypothesis that TRPM7 blocked invasion by interfering with the phosphorylation cascade induced by EBA-175 RII was tested by performing a global phosphoproteomic analysis. Erythrocytes were treated with EBA-175 RII in the absence or presence of FTY720, labeled at protein primary amine residues with a heavy (H) or an intermediate (M) dimethyl isotope respectively, and mixed with light (L) dimethyl labeled untreated erythrocytes (1:1:1). Average H/L ratios of unique peptides singly, doubly or triply phosphorylated of EBA-175 RII-treated erythrocytes were higher than average M/L ratios in EBA-175 RII/FTY720 host cells, indicating that FTY720 inhibited the phosphorylation cascade activated by EBA-175 RII binding to GPA (*Figure 4A*). Importantly, quantitative analysis of confidently identified phosphoproteins (peptide FDR < 1%) revealed a significant downregulation of phosphorylation of host cytoskeletal proteins when treated with EBA-175 RII in the presence of FTY720 ($p < 0.05$), in particular GPA (*Figure 4B*). Therefore, FTY720, a TRPM7 inhibitor, blocked phosphorylation of erythrocyte proteins induced by EBA-175 RII binding to GPA. Consequently, FTY720 inhibits merozoite invasion by blocking the phosphorylation cascade activated by EBA-175 RII binding to GPA in the erythrocyte.

Physiological shear stress in the circulation causes a reversible increase in erythrocyte Ca$^{2+}$ permeability (*Larsen et al., 1981*). In a similar manner, the EBA-175 RII induced increase in deformability could increase erythrocyte calcium permeability. A high content fluorescence imaging method was established and used to test Ca$^{2+}$ uptake into Fluo-4-AM (a Ca$^{2+}$ indicator) loaded erythrocytes

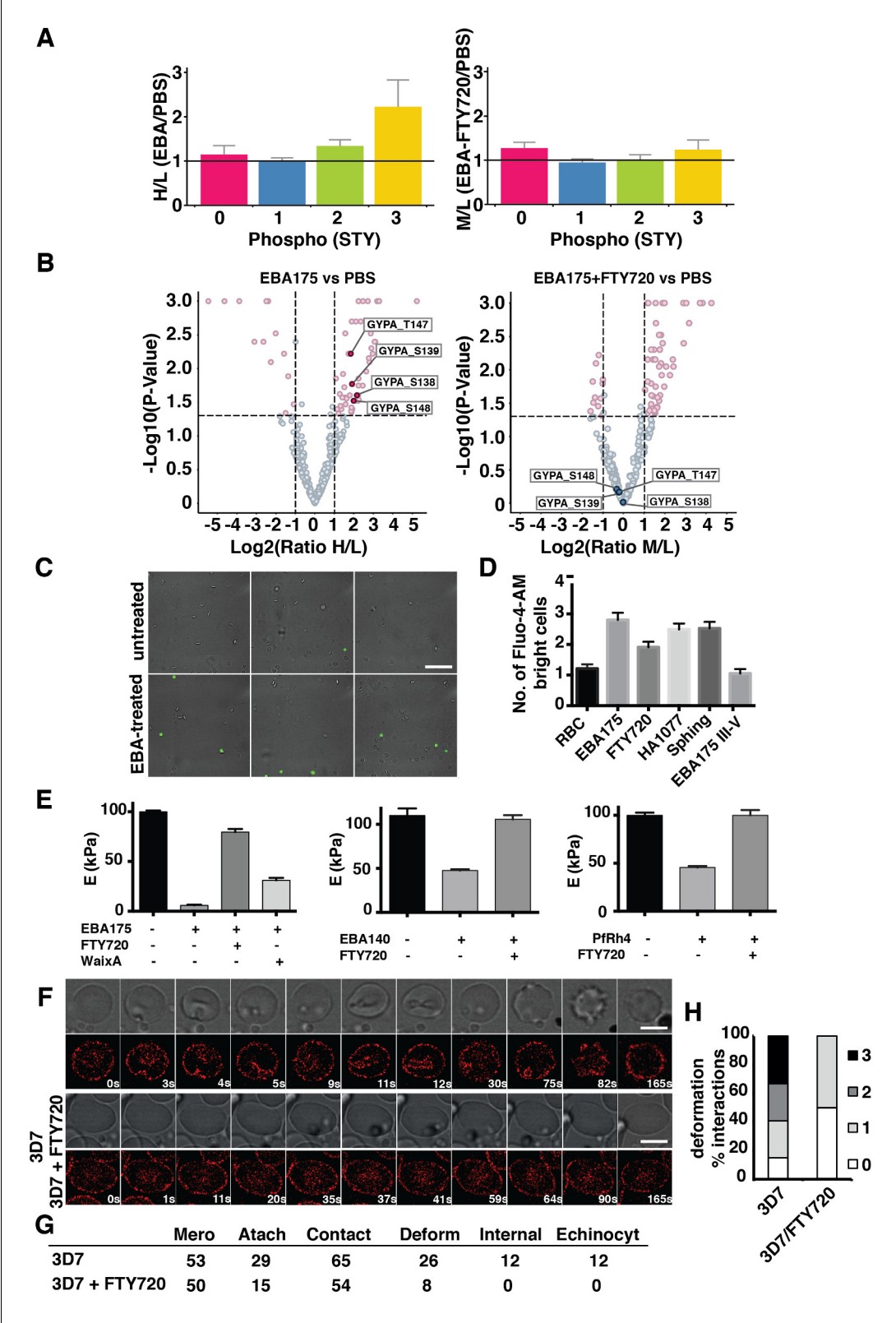

**Figure 4.** FTY720 inhibits *P. falciparum* invasion by interfering with a post-translational pathway triggered by EBA-175 binding to GPA. (**A**) Phosphopeptide heavy (H) to light (L) ratios of EBA-175 (H) with PBS (L) treated ghosts (left). Phosphopeptide intermediate (M) to light (L) ratios of EBA-175 and FTY720 treated (M) with untreated (L) ghosts (right). (**B**) Equivalent quantitative analysis of unique phosphopeptides represented in volcano plots showing that GPA phosphorylation sites are not phosphorylated in the presence of FTY720. (**C**) Fluorescence microscopy showing of erythrocytes

*Figure 4 continued on next page*

*Figure 4 continued*

in the presence of calcium indicator Fluo-4-AM. EBA-175 treated erythrocytes (bottom) are more permeable to $Ca^{2+}$. Scale bar 40 μm. (D) High content screen analysis of the number of Fluo-4-AM bright cells untreated or treated with EBA-175 RIII-IV and EBA-175 RII in the presence of FTY720, HA1077 and sphingosine (50 μM). (E) AFM Young's modulus measurements. Erythrocytes treated with PBS or EBA-175 RII in the presence of FTY720 or waixenicin A (left). Erythrocytes treated with EBA-140 RII (middle) or PfRh4 (right) in the absence or presence of FTY720. (F) Frames of live imaging experiments for 3D7 merozoites and erythrocytes in the absence or presence of FTY720 (6 μM). In each condition, bright field is shown in top panels and 594 channels in the bottom ones. Scale bars are 5 μm. (G) Table detailing the number of merozoites (Mero) that attach, contact (number of contacted erythrocytes), deform, invade and undergo echinocytosis (Echinocyt). (H) Stacked graphs showing the deformation score percentage of total interactions for 3D7 merozoites in the absence or presence of FTY720. Deformation scores are according to a simplified four-point deformation scale (*Weiss et al., 2015*). Error bars show SEM corresponding to three independent experiments.

incubated with EBA-175 RII (*Figure 4C*). Imaging showed an increment in the number of fluorescent cells when incubated with EBA-175 RII suggesting either a decrease in the integrity of the cell membrane due to increased deformability or a directly induced $Ca^{2+}$ uptake (*Figure 4D*). Additionally, inclusion of FTY720 with EBA-175 RII decreased the number of fluorescent cells detected (p<0.03) whereas HA1077 and sphingosine did not show a significant effect (p>0.05). Furthermore, FTY720 and waixenicin A inhibited the increased deformability caused by binding of EBA-175 RII to GPA as measured by AFM (p<0.0001) (*Figure 4E*), confirming the causal connection between deformability and the changes in phosphorylation upon EBA-175 RII binding. AFM also showed that FTY720 inhibited the changes in erythrocyte stiffness caused by binding of EBA-140 RII and PfRh4 (p<0.001) (*Figure 4E*) consistent with invasion ligands other than EBA-175 RII having the same effect on the TRPM7 dependent phosphorylation pathway.

To determine if FTY720 decreased the ability of merozoites to deform the erythrocyte during invasion, schizonts were purified and added to erythrocytes stained with Bodipy TR Ceramide, a viable membrane dye that allows invasion (*Volz et al., 2016*). Live imaging of merozoites in the process of interaction with an erythrocyte showed that, in the presence of FTY720, the parasite was able to contact and attach, but could not invade (*Figure 4F and G*, *Videos 1* and *2*). Furthermore, they were only able to weakly deform the erythrocytes with deformation scores (*Weiss et al., 2015*) of 0–1, as compared to merozoites in the absence of inhibitor, which were able to dramatically deform the host cell (*Figure 4G and H*). Overall, merozoites in the presence of FTY720 were able to contact and attach to erythrocytes as well as untreated parasites but showed either weak or no deformation of the cell they attempted to invade (*Figure 4H*). The decrease or lack of deformation by treated parasites confirmed that FTY720 blocked merozoite invasion by decreasing the deformability of the erythrocyte as a result of inhibition of the phosphorylation cascade induced by interaction of parasite ligands with host receptors.

## Discussion

Our results showed that interaction of *P. falciparum* merozoites with the outside of the erythrocyte activates a phosphorylation cascade

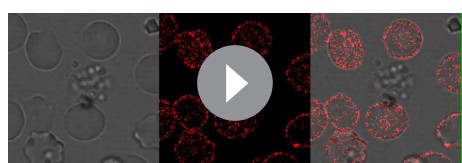

**Video 1.** This video is linked to *Figure 4F* which shows the still time lapse images of this video-microscopy of *P. falciparum* merozoites invading Bodipy TR Ceramide labelled human erythrocytes. This shows the deformation caused by the merozoites during the initial interaction before invasion is activated.

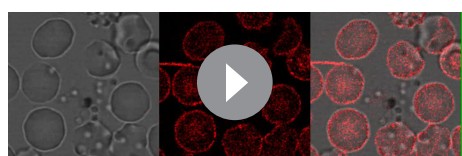

**Video 2.** This video is linked to *Figure 4F* which shows the still time lapse images of this video-microscopy of *P. falciparum* merozoites attempting to invade Bodipy TR Ceramide labelled human erythrocytes in the presence of FTY720. This shows the lack of severe deformation caused by the merozoites during the initial interaction and their inability to invade. This is linked to *Figure 4F* which shows the still time lapse images of the movie.

resulting in increased deformability and that this activation is essential for successful invasion. Whilst we have used soluble EBA-175 RII to bind to GPA over the surface of the erythrocyte the merozoite only interacts with a small portion of the surface, and it is likely that the increased deformability is more local and that the interaction of the EBA and PfRh protein families conditions this area so that the parasite can more easily deform the host cell, a step that is required for subsequent tight junction formation and entry (*Weiss et al., 2015*). Whilst this step appears to be essential EBA-175 function is generally redundant (*Duraisingh et al., 2003*; *Lopaticki et al., 2011*) but the fact that FTY720 and waixenicin A block invasion in the absence of this ligand suggests phosphorylation is a required step in invasion. The ability of other members of the EBA and PfRh family of proteins to render the erythrocyte more deformable suggests they are capable of performing this function to condition areas of the host cell as well as signaling downstream events in the invasion pathway (*Tham et al., 2015*).

Interaction of EBA-175 with the erythrocyte surface compromises the mechanical stability of the membrane and its skeleton. Many constituent proteins of the erythrocyte membrane skeleton can be phosphorylated by various kinases, and phosphorylation of $\beta$-spectrin by casein kinase and protein 4.1R by PKC has been documented to modulate the erythrocyte membrane mechanical stability (*Manno et al., 2005*). A recent study showed that activation of endogenous PKA by cAMP decreases membrane mechanical stability and that this effect is mediated primarily by phosphorylation of dematin (*Koshino et al., 2012*). Our data show that EBA-175 binding to GPA triggers the phosphorylation of trans-membrane spanning and cytoskeletal proteins, including GPA, a mechanism that weakens the erythrocyte membrane and increases cytosolic calcium concentration, as a result of permeability rather than receptor mediated store release as erythrocytes lack intracellular stores (*Cahalan et al., 2015*). FTY720, a transient receptor potential cation channel (TRPM7) inhibitor, reduces quantitatively the level of phosphorylation on the erythrocyte membrane and cytoskeleton, induced by EBA-175. TRPM7 plays a key role in the regulation of calcium and magnesium homeostasis (*Langeslag et al., 2007*), and although the function of TRPM7 kinase domain (*Ryazanova et al., 2004*) in human cells is still unclear, it autophosphorylates, autoregulates and it can also phosphorylate other proteins such as tropomodulin-1, $\beta$-actin (*Dorovkov et al., 2008*), annexin I (*Dorovkov and Ryazanov, 2004*) and myosin IIA (*Clark et al., 2006*). FTY720 has been shown to suppress TRPM7-dependent motility of HEK293 cells (*Chokshi et al., 2012*) and to abrogate erythrocyte rigidity in trauma/haemorrhagic shock (*Bonitz et al., 2014*).

This study shows that inhibitors of TRPM7 block parasite invasion by interfering with host phosphorylation mechanisms making the area around which the merozoite is interacting less deformable. Therefore the initial interaction of parasite ligands with erythrocyte receptors activates host cell changes that are essential for successful invasion and infection.

## Materials and methods

### Reagents

3D7 is a cloned line derived from NF54, supplied by David Walliker, Edinburgh University. Disruption of EBA-175 was previously published (*Lopaticki et al., 2011*). Purified recombinant EBA-175 RII antigen was kindly provided by Science Applications International Corporation (SAIC). Sequence encoding EBA-175 RIII-V was codon-optimized for expression in *E. coli* and synthesized by Genscript. Antibodies against EBA-175 RII were obtained as described before (*Healer et al., 2013*). Recombinant PfRh4 containing the full erythrocyte-binding domain (Rh4.9) was obtained as described (*Tham et al., 2009*). MSPDBL1 and MSPDBL2 was expressed recombinantly in *E. coli* as previously described (*Lin et al., 2014*). EBA-140 RII was expressed and purified as described previously (*Malpede et al., 2013*; *Lin et al., 2012*). Briefly, EBA-140 RII was expressed in *E. coli* and recovered from inclusion bodies. The denatured protein (100 mg/L) was refolded by rapid dilution for 48 hr at 4°C in 50 mM Tris (pH 8.0), 10 mM EDTA, 200 mM arginine, 0.1 mm PMSF, 2 mM reduced glutathione and 0.2 mM oxidized glutathione. After refolding, EBA-140 RII was purified by ion-exchange and size exclusion chromatography into the final buffer of 10 mM HEPES pH 7.4 150 mM NaCl. Kinase inhibitors were obtained commercially. FTY720, NS8593, Y-27632, simvastatin, HA1077-fasudil, forskolin, dorsomorphin, IBMX, BAY-3606, verapamil and gadolinium (III) from Sigma.

Sphingosine, ruthenium red and GsMTx-4 from Abcam and Gö9676 from Cell Signalling. Waixenicin A was extracted and purified from soft coral tissue as described previously (*Zierler et al., 2011*).

## Erythrocyte binding assays

The erythrocyte binding assays prior to AFM indentation and rheology were performed as follows. 10 μL washed packed blood (Australian Red Cross) was mixed with different amounts of protein (1–4 μg), resuspended in 500 μL RPMI-HEPES (2% haematocrit) and incubated at room temperature for 1 hr. For controls and inhibitor tests, antibodies (10 μg/mL) and inhibitors (50 μM) were added together with EBA-175 RII. Waixenicin A was incubated and washed prior to adding EBA-175 RII. Erythrocyte binding assays to generate binding curves using EBA175 RII, Rh4.9 and Rh5 were performed as above. Erythrocytes and proteins were centrifuged for 30 s at 13000 rpm through 400 μl dibutyl phthalate (Sigma) to remove unbound protein. After aspiration of dibutyl phthalate and buffer, 10 μl 1.6 M NaCl solution was added to the erythrocyte pellet to elute bound protein. Samples were vortexed for 5 s at 3 min intervals over 10 min to ensure equal mixing of 1.6 M NaCl solution. Samples were spun at 13000 rpm, 4 min and the supernatant, containing eluted protein, was suspended in 2X SDS sample buffer. In all binding assays, a buffer control was included containing RMPI wash buffer + amount of protein suspension buffer equivalent to the highest concentration of protein used. In one repeat experiment binding was performed in PBS instead of RMPI wash buffer.

## Western blotting

Samples were denatured in 2X SDS reducing sample buffer and loaded on 4–12% or 10% Bis-Tris or 3–8% Tris-Acetate gels (Invitrogen). Gels were run at 150V in MOPS or Tris-Acetate for one hour, and electrophoresed proteins were transferred to nitrocellulose membrane (Protran, Whatman) via wet transfer protocol. Blocking solution used was skim milk solution or Odyssey blocking buffer (LiCor). Protein bands after incubation with HRP-conjugated secondary antibody were detected by ECL detection reagent (GE Healthcare) and exposure to X-ray film (FUJIFILM). In the case of fluorescent WB, visualisation and quantitation was carried out with the LiCor Odyssey scanner (800 nm and 700 nm wavelength channels) and Odyssey software.

## Densitometry quantitation

Fluorescently probed membranes were scanned and analysed by LiCor Odyssey imaging software. An extra-sum-of-squares F test was used to compare the linear regression of the Log (protein levels) and Log (adhesin amount) to a line where y-intercept and slope = 0.

## Flow cytometry quantitation

Erythrocyte binding assay was adapted from *Salinas et al. (2014)* (*Tolia et al., 2005*). Briefly, 1 uM RII-140-6xhis, purified identically as untagged RII-140, was added to the RBCs for 1 hr at room temperature and then washed three times with DMEM 10% FCS. Anti-his antibody conjugated to FITC was then incubated with the samples for 1 hr at room temperature and then washed three times with DMEM 10% FCS. Finally, RBCs were diluted to 2 mL in DMEM 10% FCS. Samples, in triplicate, were run using a BD FACSCanto machine to collect 30,000 counts each and analyzed with FlowJo software.

## Neuraminidase treatment of erythrocytes

50 μL washed packed blood (Australian Red Cross) was mixed with 10 μL Neuraminidase from *Vibrio cholerae* (Sigma), 90 μL RPMI-Hepes/NaHCO$_3$ and incubated in a waterbath at 37°C for 1 hr. Treated cells were spun down and washed five times with RPMI-Hepes/NaHCO$_3$ before incubation for binding assays and AFM measurements.

## Atomic force microscopy indentation of erythrocytes

Erythrocyte solutions were deposited onto freshly cleaved mica and an MFP3D-BIO instrument (Asylum Research) used. Samples were nanoindented under RPMI-HEPES with MLCT silicon nitride probes (nominal spring constant 0.1 N/m, resonant frequency 38 kHz, Bruker AFM Probes). To calculate erythrocyte modulus a total of 1000 force curves were registered distributed in at least 20 different points for each condition. Analysis was made by NanomechPro software (Asylum Research).

## Rheology measurements

Samples were centrifuged at 6000 rpm for 1 min and the 10 µL blood pellet resuspended in 600 µL polyvinylpyrrolidone (PVP) solution at 25 mPa second viscosity (Rheo Meditech). Elongation index (EI) over 1–20 Pa shear stress was measured three times for each sample in a RheoScan rheometer according to manufacturer's instructions (RheoMeditech, Seoul, Korea). EI at low shear stress (3 Pa) was plotted for every concentration of EBA-175 RII. At low shear stress rheology presents the highest sensitivity to RBC membrane properties, and erythrocytes exhibit maximal deformability (*Musielak, 2009*).

## Ghost preparation and radiolabeling

Erythrocytes were washed and resuspended (50% haematocrit) with calcium-free Krebs-Ringer buffer (Sigma). 125 µL of cell suspension was incubated for 2 hr with 125 µCi of $^{32}P$ (Perkin Elmer) and either 18 µg of EBA-175 RII or the equivalent volume of Krebs-Ringer buffer. Suspensions were spun down at 3000 g for 4 min at room temperature and the supernatant was discarded. Erythrocyte pellets were lysed with ice cold 5 mM sodium phosphate pH eight buffer with protease (cOmplete, Sigma) and phosphatase (Halt, Thermo Fisher) inhibitors, spun down at 100000 g and washed three times with the same buffer. The resulting ghost pellets were snap-frozen in liquid Nitrogen and stored at −80°C until further use.

## 2D gel electrophoresis

Ghost pellets were subjected to a series of freezing/thawing cycles to remove remaining haemoglobin and resuspended in 300 µL of 2-DE sample buffer (7 M Urea, 2M Thiourea, 4% CHAPS, 50 mM DTT, 1% ampholytes), loaded onto 13 cm pI 4–7 IPG strips by passive rehydration and focused at a current limit of 50 mA/IPG strip using a fast voltage gradient (8000 V max, 24,000 Vh) at 15°C. The second dimension was carried out on pre-cast 4–12% Bis-Tris Midi Protein Gels using a NuPAGE Novex system (Life Technologies) at 75 V constant voltage and 10°C. Analytical 2-D gels were transferred to PVDF membranes using an iBlot Dry Blotting System (Life Technologies) and imaged using BAS-IP SR 2040 E phosphor storage plates (GE Healthcare) for 48 hr. High-resolution imaging and accurate quantitation of $^{32}P$-labeled protein spots was achieved using a Typhoon FLA 7000 laser-scanning detection system (GE Healthcare). Preparative 2-DE gels were stained using Colloidal Coomassie Brilliant Blue (Sigma) and spots matching $^{32}P$-labeled protein spots manually excised and subjected to LC-MS/MS analysis.

## Gel excision, in-gel digestion and nano-LC-MS/MS

Protein spots were manually excised from preparative 2-D gels and subjected to manual in-gel reduction, alkylation and tryptic digestion. All gel samples were reduced with 10 mM DTT (Sigma) for 30 min, alkylated for 30 min with 50 mM iodoacetamide (Sigma) and digested with 375 ng trypsin gold (Promega) for 16 hr at 37°C. The extracted peptide solutions were then acidified (0.1% formic acid) and concentrated to 10 mL by centrifugal lyophilisation using a SpeedVac AES 1010 (Savant). Extracted peptides were injected and fractionated by reversed-phase liquid chromatography on a nanoACQUITY UHPLC system (Waters, USA) using a nanoACQUITY C18 150 mm × 0.15 mm I.D. column (Waters, USA) developed with a linear 60 min gradient with a flow rate of 250 nL/min from 100% solvent A (0.1% formic acid in Milli-Q water) to 60% solvent B (0.1% formic acid, 60% acetonitrile (Thermo Fisher, USA), 40% Milli-Q water). The nano-UHPLC was coupled on-line to a Q-Exactive Orbitrap mass spectrometer equipped with a Proxeon nano-electron spray ionization source (Thermo Fisher, USA) for automated MS/MS. High mass-accuracy MS data was obtained in a data-dependent acquisition mode with the Orbitrap resolution set at 75,000 and the top-ten multiply charged species selected for fragmentation by HCD (single and doubly charged species were ignored). The ion threshold was set to 15,000 counts for MS/MS. The CE voltage was set to 27.

## Mass spectra database searching

For protein identification of protein spots LC-MS/MS data were searched against a non-redundant protein decoy database comprising sequences from *P. falciparum*, as well as their reverse sequences and common contaminants. Mass spectra peak lists were extracted using extract-msn as part of Bioworks 3.3.1 (Thermo Fisher Scientific) linked into Mascot Daemon (Matrix Science, UK). Peak lists for

each nano-LC-MS/MS run were searched using MASCOT v2.2.04 (Matrix Science, UK), provided by the Australian Proteomics Computational Facility (www.apcf.edu.au). The search parameters consisted of carbamidomethylation of cysteine as a fixed modification (+57 Da), with variable modifications set for $NH_2$-terminal acetylation (+42 Da) and oxidation of methionine (+16 Da). A precursor mass tolerance of 20 ppm, fragment ion mass tolerance of ±0.04 Da and an allowance for up to three missed cleavages for tryptic searches was used. Scaffold v4.4.3 (Proteome Software, Inc., USA) was used to validate protein identifications derived from MS/MS sequencing results (FDR <0.5%).

## Mass spectrometry sample preparation

For global phosphoproteomics analyses, ~200 µG of frozen ghost pellets were extracted in 200 µL lysis buffer (1% SDS, 2 x Complete Protease Inhibitor Cocktail-EDTA (Roche), in 10 mM HEPES pH 8.5), reduced with 20 mM TCEP (Sigma) for 15 min, alkylated for 30 min with 50 mM iodoacetamide (Sigma) and digested with 4 µG trypsin gold (Promega) for 16 hr at 37°C, using the SP3 on-bead digestion method (*Hughes et al., 2014*). The extracted peptide solutions were then desalted using C18 MacroSpin columns and on-column stable isotope dimethyl labeling of differentially treated samples was performed using light ($^{12}CH_2O$), medium ($^{12}CD_2O$) or heavy ($^{13}CD_2O$) formaldehyde (Sigma), as previously described (*Boersema et al., 2009*). Differentially labelled peptide samples were then mixed 1:1 and phosphopeptide enrichment was achieved using sequential elution from IMAC or TiO2 columns, as detailed elsewhere (*Thingholm et al., 2008*).

## Quantitative proteomics analysis

Relative quantitation of differentially labeled RBC phosphopeptide preparations was performed using the MaxQuant software package (*Cox and Mann, 2008*). High-resolution MS data were searched using a tolerance of 10 ppm for precursor ions and 20 mmu for productions. Enzyme specificity was tryptic and allowed for up to two missed cleavages per peptide. Carbamidomethylation of cysteines (+57) was specified as a constant modification with oxidation of methionine (+16), protein N-terminal acetylation (+52) and light (L; $\triangle$ = +28), intermediate (M; $\triangle$ = +32) or heavy (H; $\triangle$ = +36) dimethyl labels on the peptide N termini and lysine residues set as variable modifications. The MS data were searched against human, bovine, or *P. falciparum* proteins in the non-redundant MSPnr protein database at a 1% false discovery rate (FDR). A robust permutation test was used to analyze MaxQuant data and evaluate statistically significant differences in the relative abundance of RBC phosphopeptides (*Nguyen et al., 2012*).

## Inhibitor screen

Highly synchronous mature schizonts (from a 30 mL culture of >5% parasitemia) were magnet purified (MACS; Miltenyi Biotec) and plated in triplicates (50 µL) at 1% parasitemia and 1% haematocrit. Parasites were allowed to egress and reinvade fresh erythrocytes for a period of 12–16 hr at 37°C in the presence of inhibitors (50 µM) or equivalent volumes of buffer (water, DMSO or KR buffer). Parasitemia was determined by GIEMSA staining.

## Merozoite invasion inhibition assays

Merozoites were purified based on an established method (*Boyle et al., 2010*). Highly synchronous mature schizonts were isolated from uninfected erythrocytes with a MACS magnet separation column (Miltenyi Biotec). Purified schizonts were incubated with 10 µM of the cysteine protease inhibitor E64 to prevent schizont ruptures. After 5–6 hr of incubation, schizont pellets were passed through a 1.2 µm syringe filter (Acrodisc; 32 mm; Pall). Filtrate containing purified merozoites was immediately added to fresh erythrocytes (70–80% hematocrit). The mix was transferred to a 96 well round bottom microtitre plates (Falcon) in 50 µL aliquots with doubling dilutions of each inhibitor (5–250 µM) or control buffer and incubated in a shaker at 37°C. After incubation for 24 hr each well was fixed at RT for 30 min with 50 µL of 0.25% glutaraldehyde (ProSciTech) diluted in PBS. Following centrifugation at 1200 rpm for 2 min, the supernatants were discarded and trophozoite stage parasites were stained with 50 mL of 5X SYBR Green (Invitrogen) diluted in PBS. The parasitemia of each well was determined by counting 50,000 cells by flow cytometry using a Cell Lab Quanta SC – MPL Flow Cytometer (Beckman Coulter). Invasion rate was calculated as %-invaded relative to untreated controls. All samples were tested in triplicate.

## Growth inhibition assays.

A Growth Inhibition Assay (GIA) protocol was modified from a previously described method (*Persson et al., 2006*). Trophozoite stage parasites at 0.5% parasitemia were grown in a 50 µL culture at 2% hematocrit in 96 well round bottom microtitre plates (Falcon) with doubling dilutions of each each inhibitor (5–250 µM) or control buffer. After incubation for 48 hr each well was fixed at RT for 30 min with 50 µL of 0.25% glutaraldehyde (ProSciTech) diluted in PBS. Following centrifugation at 1200 rpm for 2 min, the supernatants were discarded and trophozoite stage parasites were stained with 50 mL of 5X SYBR Green (Invitrogen) diluted in PBS. The parasitemia of each well was determined by counting 50,000 cells by flow cytometry using a Cell Lab Quanta SC – MPL Flow Cytometer (Beckman Coulter). Growth was expressed as a percentage of the parasitemia obtained using an inhibitor-free control. All samples were tested in triplicate.

## High content fluorescence screen

Erythrocyte binding assays in this case were performed as follows. Erythrocytes (Australian Red Cross) were previously incubated with Fluo-4 AM (Life Technologies), 1:250 dilution to 1% haematocrit erythrocytes in RPMI-HEPES and washed. 2.5 µL packed RBCs were mixed with EBA-175 RII or RIII-V (2 µg), resuspended in 50 µL RPMI-HEPES (5% haematocrit) and incubated at RT for 1 hr. FTY720 and sphingosine were added together with EBA-175 RII at a 50 µM concentration. After incubation samples were diluted with RPMI-HEPES to 0.05% haematocrit and transferred to 100 µL aliquots into BD Falcon 384 well TC-treated plates. Plates were read in an Opera Phenix high content screening system for brightfield, phase contrast and 488 nm fluorescence. These experiments were repeated three independent times and standard error of the mean calculated.

## Live imaging of merozoite invasion

Fresh erythrocytes (1% hematocrit) were washed in complete RPMI-Hepes culture medium. Bodipy TR Ceramide (Thermo Fisher Scientific) was added at 1:1000 dilution and incubated for >1 hr. Erythrocytes were washed 2–3 times with complete RPMI-Hepes and then resuspended in 2 mL of complete RPMI-Hepes. Highly synchronous schizonts (>5% parasitemia from a ~3% hematocrit 30 mL culture) were magnet purified, then added to the labelled erythrocytes (0.1% haematocrit) and 2 mL of this transferred to a 35 mm Fluorodish (World Precision Instruments). Live imaging was performed at 37°C on a Leica SP8 confocal microscope. A 63x/1.4 NA Oil Immersion objective on the Leica SP8 confocal. Time-lapsed images were collected every second upon schizont rupture (594 filter) with an 8 kHz resonant scanner with 4x line averaging and HyD detectors. Cells were maintained at 37°C in a low $O_2$ and $CO_2$ nitrogen atmosphere. ImageJ Fiji was used to assemble image series and perform image analyses. Deformation scores were determined according to an established method (*Weiss et al., 2015*).

## Acknowledgements

We would like to thank Matthew WA Dixon for helpful discussions and assistance with the ektacytometry experiments. We also thank Leann Tilley and University of Melbourne for access to the ektacytometer. We thank Australian Red Cross Blood Service for blood.

## Additional information

### Funding

| Funder | Grant reference number | Author |
|---|---|---|
| Howard Hughes Medical Institute | HHMI International Scholar Award 55007645 | Alan F Cowman |
| National Health and Medical Research Council | NHMRC Program Grant 637406 | Alan F Cowman |
| Australian Research Council | Australian Research Council Future Fellowship | Wai-Hong Tham |
| National Health and Medical | IRIISS grant | Xavier Sisquella |

| Research Council | | Thomas Nebl |
| | | Jennifer K Thompson |
| | | Lachlan Whitehead |
| | | Kelly Rogers |
| | | Joseph O'Neill |
| | | Wai-Hong Tham |
| | | Alan F Cowman |
| National Health and Medical Research Council | 1026581 | Alan F Cowman |

The funders had no role in study design, data collection and interpretation, or the decision to submit the work for publication.

## Author contributions

XS, Conceptualization, Data curation, Formal analysis, Investigation, Visualization, Methodology, Writing—original draft, Writing—review and editing; TN, Conceptualization, Data curation, Formal analysis, Validation, Investigation, Visualization, Methodology, Writing—review and editing; JKT, JO'N, Data curation, Investigation, Methodology, Writing—review and editing; LW, KR, Conceptualization, Data curation, Formal analysis, Investigation, Methodology, Writing—review and editing; BMM, NDS, NHT, AF, FDH, Resources, Methodology, Writing—review and editing; W-HT, Conceptualization, Data curation, Supervision, Investigation, Methodology, Writing—review and editing; AFC, Conceptualization, Data curation, Formal analysis, Supervision, Funding acquisition, Writing—original draft, Project administration, Writing—review and editing

## Author ORCIDs

Niraj H Tolia, http://orcid.org/0000-0002-2689-1337
Alan F Cowman, http://orcid.org/0000-0001-5145-9004

# Additional files

## Supplementary files

• Supplementary file 1. Table of LC-MSMS evidence and quantitative statistics for the double labelling experiment in which EBA-175 treated erythrocyte proteins were labelled with heavy isotope (H) versus untreated that were labelled with light isotope (L).

• Supplementary file 2. Table of LC-MSMS evidence and quantitative statistics for the triple labelling experiment in which EBA-175 treated erythrocytes were labelled with heavy isotope (H), EBA-175 +FTY720 treated erythrocyte proteins were labelled with medium isotope (M) versus untreated erythrocyte proteins labelled with light isotope (L).

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
