## [Decision Letter]

Thank you for submitting your article "*P. falciparum* ligand binding to erythrocytes induce alterations in deformability essential for invasion" for consideration by *eLife*. Your article has been favorably evaluated by Wenhui Li (Senior Editor) and three reviewers, one of whom is a member of our Board of Reviewing Editors. The reviewers have opted to remain anonymous.

The reviewers have discussed the reviews with one another and the Reviewing Editor has drafted this decision to help you prepare a revised submission.

Summary:

It has been known for more than 20 years that EBA175 binding to GYPA is a critical interaction for *P. falciparum* erythrocyte invasion. This manuscript begins to clarify the molecular mechanism. It describes experiments that implicate a phosphorylation cascade, activated by binding of *P. falciparum* EBA-175 to glycophorin, in determining an enhanced deformability of the red blood cell surface necessary for merozoite invasion. In particular, the manuscript provides data showing (1) increased global deformability when EBA-175 binds; (2) increased phosphorylation of rbs cytoskeletal proteins following EBA-175 ligation of glycophorin A; (3) inhibition of parasite growth by certain kinase inhibitors and identification of TRPM7, which targets tropomodulin-1, as a likely candidate member of the implied kinase cascade; (4) inhibition of EBA-175 induced cytoskeletal phosphorylation by the same kinase inhibitors(s); (5) inhibition of merozoite-induced, local deformability by an inhibitor of TRPM7.

Essential revisions:

1) Controls. A question throughout the manuscript is whether the observed effects are the direct result of EBA175 binding. This assumption underlies all the major conclusions, but it needs more complete proof. A panel of alternative ligands is used in Figure 1 showing differential effects, but in all subsequent experiments, the conditions that are being compared are either EBA175 in the presence and absence of anti-EBA175 antibodies (Figure 1), or EBA175 binding vs. PBS (for the proteomics). These experiments are simply not carefully controlled enough to ascertain that it is the act of EBA175 binding to GYPA that is causing the observed phenomenon (both effect on erythrocyte stiffness, and effect on erythrocyte protein phosphorylation). Given that the structure of the EBA175-GYPA complex is known, and there has been extensive site-directed mutagenesis of EBA175, it would be a relatively simple matter to make a version of recombinant EBA175 that is unable to bind to GYPA and to use this protein as a control for both stiffness and phosphorylation assays (not necessarily global phosphoproteomics, but immunoblots, or targeted triple-quad of key phosphotargets such as GYPA). These comparisons would show unequivocally that it is EBA175 binding that is important, not a non-specific effect. Other controls are also possible, such as using neuraminidase treated ghosts to eliminate EBA175 binding – anything that actually convincingly shows that the effects are direct effect of EBA175 binding to GYPA.

2) Amount of EBA175 being added. While the effects are marked and interesting, it is not clear whether the amount of recombinant protein being added is in any way physiological, or how it relates to the amount of GYPA on the surface of erythrocytes. Micrograms is not the most useful scale (as in Figure 1); μm would be more standard. It is also important to know how this value, whatever it is, convert to numbers of EBA175 dimers and hence allow comparison with the number of GYPA molecules on an erythrocyte surface. Does 2.5μg equate to a level at which they would expect all GYPA molecules to be saturated?

3) Do other ligands have similar effects – is the result generalizable? While binding data in Figure 1 show an effect for only EBA175, waixenicin inhibits a EBA175 knockout strain with a similar efficiency to wild-type strains, suggesting that other ligand interactions may also use the same TRPM7 dependent pathway. The lab has a wide array of ligand knockout *P. falciparum* strains, and some of these may be more revealing to try to tease out the generalizable nature of the phenomenon – are they all affected by TRPM7 inhibitors to the same extent, or is there some variation?

---

## [Author Response]

Essential revisions:

1) Controls. A question throughout the manuscript is whether the observed effects are the direct result of EBA175 binding. This assumption underlies all the major conclusions, but it needs more complete proof. A panel of alternative ligands is used in Figure 1 showing differential effects, but in all subsequent experiments, the conditions that are being compared are either EBA175 in the presence and absence of anti-EBA175 antibodies (Figure 1), or EBA175 binding vs. PBS (for the proteomics). These experiments are simply not carefully controlled enough to ascertain that it is the act of EBA175 binding to GYPA that is causing the observed phenomenon (both effect on erythrocyte stiffness, and effect on erythrocyte protein phosphorylation). Given that the structure of the EBA175-GYPA complex is known, and there has been extensive site-directed mutagenesis of EBA175, it would be a relatively simple matter to make a version of recombinant EBA175 that is unable to bind to GYPA and to use this protein as a control for both stiffness and phosphorylation assays (not necessarily global phosphoproteomics, but immunoblots, or targeted triple-quad of key phosphotargets such as GYPA). These comparisons would show unequivocally that it is EBA175 binding that is important, not a non-specific effect. Other controls are also possible, such as using neuraminidase treated ghosts to eliminate EBA175 binding – anything that actually convincingly shows that the effects are direct effect of EBA175 binding to GYPA.

Whilst mutation of EBA175 RII to engineer a form that does not bind GYPA would be a useful control the specific mutations to do this are not necessarily completely obvious. We believe that the suggestion to test neuraminidase-treated erythrocytes to remove sialic acids is a better control. Therefore, we have done additional AFM measurements of erythrocytes treated with neuraminidase with and without EBA175 RII. EBA-175 RII causes no deformability change on neuraminidase treated RBCs, confirming that the change of erythrocyte mechanical properties observed upon addition of EBA-175 RII on untreated RBCs is a direct consequence of EBA-175 RII binding to GYPA. An additional panel has been inserted into Figure 1 (Figure 1).

2) Amount of EBA175 being added. While the effects are marked and interesting, it is not clear whether the amount of recombinant protein being added is in any way physiological, or how it relates to the amount of GYPA on the surface of erythrocytes. Micrograms is not the most useful scale (as in Figure 1); μm would be more standard. It is also important to know how this value, whatever it is, convert to numbers of EBA175 dimers and hence allow comparison with the number of GYPA molecules on an erythrocyte surface. Does 2.5μg equate to a level at which they would expect all GYPA molecules to be saturated?

According to Poole et al., 2000, a red blood cell contains approx. 5x10^5^ GYPA sites (10^6^ copies). In a 50% hematocrit suspension of erythrocytes there is approximately 2 million cells per microliter. Our binding suspensions were 2% hematocrit and the concentration of the GYPA binding sites is approx. 30 nM. Taking into account that EBA-175 RII binds each GYPA site as a dimer, we can consider that a 50 nM concentration of EBA-175 RII (Figure 1) is saturating all GYPA binding sites. Reference to this has been inserted into the manuscript. Figure 1 axis have been expressed in nanomolar and correlated to microgram in the figure legend.

3) Do other ligands have similar effects – is the result generalizable? While binding data in Figure 1 show an effect for only EBA175, waixenicin inhibits a EBA175 knockout strain with a similar efficiency to wild-type strains, suggesting that other ligand interactions may also use the same TRPM7 dependent pathway. The lab has a wide array of ligand knockout P. falciparum strains, and some of these may be more revealing to try to tease out the generalizable nature of the phenomenon – are they all affected by TRPM7 inhibitors to the same extent, or is there some variation?

We have used AFM to show that FTY720 inhibits the effect of EBA-140 and PfRh4 on erythrocyte deformability, which confirms that these ligands also activate TRPM7 to deform red blood cells. These results have now been inserted into the manuscript in Figure 4.

In respect of the comment suggesting that additional knockout strains be tested we have of course considered this and already include the EBA-175 knockout line analysis. Analysis of additional knockout strains would not be informative because the function of the EBA and Rh ligands are redundant as they have overlapping function as we have shown previously. We would get the same result as we show in the manuscript for the EBA175 knockout parasites. In contrast, the best experiment to do would be to knockout TRPM7 in the erythrocyte and this should have an effect on both invasion of *P. falciparum* and the ability of the ligands to alter deformability. We are in the process of starting this experiment to knockout TRPM7 in erythroid lines. However, we believe that this is outside the scope of this manuscript.